# Effects of Gene-Environment Interaction on Obesity among Chinese Adults Born in the Early 1960s

**DOI:** 10.3390/genes12020270

**Published:** 2021-02-13

**Authors:** Weiyan Gong, Hui Li, Chao Song, Fan Yuan, Yanning Ma, Zheng Chen, Rui Wang, Hongyun Fang, Ailing Liu

**Affiliations:** Chinese Center for Disease Control and Prevention, National Institute for Nutrition and Health, Beijing 100000, China; gongwy@ninh.chinacdc.cn (W.G.); abclihui@163.com (H.L.); songchao@ninh.chinacdc.cn (C.S.); yuanfan@ninh.chinacdc.cn (F.Y.); mayn@ninh.chinacdc.cn (Y.M.); chenzheng@ninh.chinacdc.cn (Z.C.); wangrui@ninh.chinacdc.cn (R.W.); fanghy@ninh.chinacdc.cn (H.F.)

**Keywords:** SNPs, BMI, waist circumference, obesity, central obesity, gene–environment interaction

## Abstract

The prevalence of obesity has been increasing sharply and has become a serious public health problem worldwide. Gene–environment interaction in obesity is a relatively new field, and little is known about it in Chinese adults. This study aimed to provide the effects of gene–environment interaction on obesity among Chinese adults. A stratified multistage cluster sampling method was conducted to recruit participants from 150 surveillance sites. Subjects born in 1960, 1961 and 1963 were selected. An exploratory factor analysis was used to classify the environmental factors. The interaction of single nucleotide polymorphisms (SNPs) and environmental factors on body mass index (BMI) and waist circumference were analyzed using a general linear model. A multiple logistic regression model combined with an additive model was performed to analyze the interaction between SNPs and environmental factors in obesity and central obesity. A total of 2216 subjects were included in the study (mean age, 49.7 years; male, 39.7%, female, 60.3%). Engaging in physical activity (PA) could reduce the effect of *MC4R* rs12970134 on BMI (β = −0.16kg/m^2^, *p* = 0.030), and also reduce the effect of *TRHR* rs7832552 and *BCL2* rs12454712 on waist circumference (WC). Sedentary behaviors increased the effects of SNPs on BMI and WC, and simultaneously increased the effects of *FTO* rs9939609 and *FTO* rs8050136 on obesity and central obesity. A higher socioeconomic status aggravated the influence of SNPs (including *FTO* rs9939609, *BNDF* rs11030104, etc.) on BMI and WC, and aggravated the influence of *SEC16B* rs574367 on central obesity. The *MC4R* rs12970134 association with BMI and the *FTO* rs8050136 association with central obesity appeared to be more pronounced with higher energy intake (β = 0.140 kg/m^2^, *p* = 0.049; OR = 1.77, *p* = 0.004, respectively). Engaging in PA could reduce the effects of SNPs on BMI and WC; nevertheless, a higher socioeconomic status, higher dietary energy intake and sedentary behaviors accentuated the influences of SNPs on BMI, WC, obesity and central obesity. Preventative measures for obesity should consider addressing the gene–environment interaction.

## 1. Introduction

The prevalence of obesity has tripled over the past three to four decades and has become a serious public health issue and global health challenge [1]. In 2016, at least one third of the world’s adults were suffering from overweight or obesity [2]. The prevalence of overweight and obesity was lower, but was increasing faster in developing countries than in developed countries [3]. In 2010–2012, according to the Chinese criteria of weight for adults, the prevalence of obesity among residents aged 18 years and above was 11.9%, and among children and adolescents aged between 6–17 years, it was 6.4% [4]. Obesity is in relation to an increased risk of numerous chronic diseases, such as hypertension, coronary heart disease and stroke, as well as excess mortality [3,5,6]. However, obesity has a negative impact not only on health, but also on psychology and socioeconomics [7,8,9,10,11,12]. The median of mean total annual healthcare costs increased 12% and 36% for overweight and obese individuals, respectively, compared with the individuals with a healthy weight [12]. The medical costs of obesity was about USD 150 billion, accounting for almost 10% of all medical spending in the United States [8,10]. In 2010, the economic burden of major chronic diseases caused by overweight and obesity was about USD 12.85 billion, responsible for 42.9% of the economic burden of major chronic diseases in China [13]. The serious public health issues and economic burden caused by overweight and obesity made it imperative to understand their genetic and environmental factors.

Previous studies have found that the differences in the prevalence of overweight and obesity among different ethnic groups may be related to allele frequencies of obesity; in addition, environmental factors may regulate the expression of obesity genes and increase or decrease the susceptibility of people to obesity [14]. One study found that reduced outdoor activities may increase the risk of obesity in people carrying *FTO* rs9939609-A among Kazakh school-aged children [15]. Similarly, studies among Danes and Chinese school-aged children showed that low physical activity accentuated the effect of *FTO* rs9939609 on body fat accumulation [16,17]. Another study did not find an interaction between *FTO* rs9939609 and physical activity on obesity [18]. Several studies have shown that the genetics associated with obesity appeared to be more pronounced with greater intake of high-energy foods, such as fried foods, sugar-sweetened beverages and protein [19,20,21]. The effect of genes on obesity may also interact with socioeconomic status. The British Biobank’s research showed that a low socioeconomic status would aggravate the effect of the *FTO* gene on obesity [22]. Education level, as an aspect in determining socioeconomic status, worked together with genes to influence the occurrence of obesity. For example, the HELENA study found that a low education level increased the risk of obesity caused by genes [23]. A study of the Mediterranean population found that a low education level increased the *FTO* rs9939609 risk for obesity [24].

Despite comprehensive studies conducted on the interaction of genes and the environment on obesity, little is known about the effect of the gene–environment interaction among Chinese adults. A comprehensive study that evaluates environmental factors in conjunction with genetic contributions among Chinese population is imperative. The current study aimed to explore the effect of the gene–environment interaction on body mass index (BMI), waist circumference (WC), obesity and central obesity among Chinese adults born in the early 1960s.

## 2. Materials and Methods

### 2.1. Study Design and Subjects

This study was based on the 2010–2012 China Nutrition and Health Surveillance (CNHS). CNHS was a nationally representative cross-sectional study covering all 31 provinces, autonomous regions and municipalities directly under the central government of China (except Taiwan, Hong Kong and Macao). A stratified and multistage cluster random-sampling method proportional to population was employed to conduct the survey in 150 surveillance sites, with urban and rural areas divided into four stratums, including 34 metropolis surveillance sites, 41 small to medium urban surveillance sites, 45 general rural surveillance sites and 30 poor rural surveillance sites. Six neighborhood (village) committees were sampled from each surveillance site and 75 households were sampled from each neighborhood (village) committees. Subjects born in 1960, 1961 and 1963 were selected. The exclusion criteria were incomplete information (such as lack of weight or height, waist circumference, dietary data, etc.), unqualified blood samples, failure of DNA extraction or abnormal gene detection results, and those with liver, kidney or heart diseases, or cancer. Finally, a total of 2216 subjects were included in the current study. The study protocols were approved by the Ethics Committee of NINH, China CDC (No. 2013-010). All the permanent residents in the selected households were the respondents and signed the informed consent. The surveillance content included a dietary survey, a medical physical examination, an inquiring survey and a laboratory test. The data for the current study included basic household information, individual dietary behaviors (including a 24 h dietary-inquiry survey for 3 consecutive days and weighing of household seasonings), physical activity behaviors, individual health status, height, weight and waist circumference.

### 2.2. Genotyping and SNP Selection

The genotype of 16 obesity-related single nucleotide polymorphisms (SNPs) were detected by Mass ARRAY (Agera, San Diego, CA, USA). SNPs exclusion criteria: (1) detection rate < 80%; (2) deviation from the Hardy-Weinberg equilibrium *p* < 0.001; and (3) a minor allele frequency of each SNP < 5%. Finally, a total of 12 SNPs were involved in the present study. The association of the 12 SNPs with obesity had been indicated; however, whether these SNPs interacted with environment was still unclear.

### 2.3. Definition and Standards

According to the “Criteria of weight for Chinese adults,” obesity is defined as ≥ 28.0 kg/m^2^, so obesity was defined as BMI ≥ 28.0 kg/m^2^. The central obesity was defined as WC ≥ 90 cm for males and WC ≥ 85 cm for females. Environmental factors included economic level, education level, leisure-time physical activity, transportation mode, housework time, leisure sedentary behavior, daily energy intake, etc.

### 2.4. Statistical Analysis

The exploratory factor analysis was used to classify the environmental factors. First, a Kaiser–Meyer–Olkin (KMO) test and a Bartlett’s spherical test were performed to determine whether factor analysis was suitable. Factor analysis generally can be done when KMO ≥ 0.5, and Bartlett’s spherical test when *p* < 0.05, while KMO ≥ 0.6 would be more suitable [15,25]. In this study, we adopted KMO ≥ 0.5, which was used in a previous study [15]. Second, according to the results of factor analysis, the factor with eigenvalue > 1 and cumulative contribution rate > 70% was selected as the initial common factor. After orthogonal rotation, the variable with factor load ≥ 0.50 was considered as the main component of the factor. Finally, according to the 50th percentile of the factor score, each factor was divided into two categories of variables, and the single SNPs were divided into two categories according to whether they carried risk alleles. The interaction of SNP, genetic risk score and environmental factors on BMI and waist circumference was analyzed by general linear model. The model included genetic and environmental factors, and was adjusted for age and gender. A multiple logistic regression model combined with an additive model was performed to analyze the interaction between single SNPs and environmental factors on obesity and central obesity. The odds ratio (OR) is generally used to assess the risk. Let OR (GE) be the risk of interaction of SNPs and environmental factors, OR (G) be the risk when SNPs act alone, and OR(E) be the risk when environment acts alone. When the two factors were combined, the proportion attributed to interaction was AP(G*E) = (OR(GE) − OR (G) − OR (E) + 1)/OR (GE). A two-tailed *p* < 0.05 was considered statistically significant.

## 3. Results

### 3.1. Subjects Characteristics

Basic characteristics of the study subjects are presented in Table 1. The average age of the 2216 subjects was 49.7 years. The prevalence of central obesity in males was 27.5%, which was lower than that in females (33.0%, *p* < 0.05). There also were significant differences between obesity and non-obesity, and central obesity and non-central obesity in BMI and WC (*p* < 0.001).

### 3.2. Results of the Exploratory Factor Analysis

The results of exploratory factor analysis are shown in Table 2. This analysis was conducted for 7 variables, including leisure-time physical activity, housework, transportation mode, economic level, education level, energy intake and leisure-time sedentary behavior (LTSB). The KMO test value was 0.558, and the Bartlett’s spherical test value was 690.09 (df = 21, *p* < 0.0001), so it was suitable for exploratory factor analysis. After this analysis, four common factors were finally extracted. Factor 1 (including leisure-time physical activity, housework and transportation mode) was defined as physical activity (PA). Factor 2 (including economic level and education level) was defined as socioeconomic status. Factor 3 was LTSB. Factor 4 was dietary energy intake.

### 3.3. Interaction of Genes and Environment in BMI and WC

As shown in Table 3, engaging in PA could reduce the effect of *MC4R* rs12970134 on BMI (β = −0.16 kg/m^2^, *p* = 0.030), and reduce the effect of *TRHR* rs7832552 and *BCL2* rs12454712 on WC (β = −0.426 cm, *p* = 0.044; β = −0.450 cm, *p* = 0.048, respectively). A high socioeconomic status appeared to increase the effect of most SNPs on BMI and WC. A high dietary energy intake accentuated the effect of *MC4R* rs12970134 on BMI (β = 0.140 kg/m^2^, *p* = 0.049). LTSB increased the influence of *SEC16B* rs574367 and *MC4R* rs12970134 on BMI (β = 0.140 kg/m^2^, *p* = 0.044; β = 0.214 kg/m^2^, *p* = 0.003, respectively), and increased the influence of *BNDF* rs11030104 and *MC4R* rs12970134 on WC (β = 0.459 cm, *p* = 0.041; β = 0.562 cm, *p* = 0.007, respectively).

### 3.4. Interaction of Genes and Environment in Obesity and Central Obesity

None of the SNPs was found to interact with PA in obesity or central obesity (Table 4).

Significant evidence for interaction with *SEC16B* rs574367 was seen for socioeconomic status (*p* = 0.020), with a larger effect of *SEC16B* rs574367 in high socioeconomic status (OR = 1.39, 95%CI:1.05–1.82) on central obesity. When a high socioeconomic status and *SEC16B* rs574367 coexisted, the incidence of obesity was attributable to the interaction ratio of 2.74%. No interaction was found between socioeconomic status and any SNP’s effect on obesity (Table 5).

A significant interaction was found between dietary energy intake and *FTO* rs8050136 (*p* = 0.004), in which participants with a higher dietary energy intake had a more obvious effect of *FTO* rs8050136 on obesity compared to those with a lower dietary energy intake (OR = 1.77, 95%CI:1.20–2.62). The proportion of obesity attributed to this interaction was 19.84%. No interaction was found between dietary energy intake and any SNP’s effect on central obesity (Table 6).

LTSB interacted together with SNPs on obesity and central obesity. *FTO* rs9939609’s association with obesity and central obesity appeared to be more pronounced with a long-time LTSB (OR = 1.63, 95%CI:1.09–2.45; OR = 1.49, 95%CI:1.09–2.02). Interaction accounted for 2.88% and 21.62% of the occurrence of obesity and central obesity when a long-time LTSB existed with *FTO* rs9939609. Interaction with obesity and central obesity was also observed between *FTO* rs8050136 and LTSB. A long-time LTSB accentuated the effect of *FTO* rs8050136 on obesity and central obesity (OR = 1.27, 95%CI:1.05–2.36; OR = 1.44, 95%CI:1.06−1.97). When the two factors existed together, the proportion attributed to interaction was 1.59% and 20.82%, respectively. Significant interaction was also identified between LTSB and *SEC16B* rs574367 (*p* = 0.005). A higher effect on central obesity for *SEC16B* rs574367 was observed in participants with a long-time LTSB (OR = 1.39, 95%CI:1.06−1.97). The proportion attributable to the interaction was 1.23% when both factors were present (Table 7).

## 4. Discussion

The occurrence of obesity was affected by genetic and environmental factors and their interactions. The polymorphism of obesity genes was different in different races; in addition to being related to heredity, the environmental factors may also affect the gene expression. In the present study, we analyzed the interaction between obesity-related genes and environmental factors. Several environmental factors were identified that influence the effect of SNPs on BMI, WC, obesity and central obesity.

Many studies have shown that regular PA can reduce the effect of genes on obesity [16,19,20,21]. In contrast, the genetic association with BMI was accentuated with increasing prolonged television-watching [26]. A study conducted among children in Beijing found that the association between *BDNF* rs6265 and obesity risk was only identified in children with moderate to low levels of PA or sedentary behavior [27]. Consistent with the above results, our results indicated that the interaction associated with PA attenuated the effect of *MC4R* rs12970134 on BMI, and the effect of *TRHR* rs7832552 and *BCL2* rs12454712 on WC. LTSB increased the effects of *SEC16B* rs574367 and *MC4R* rs12970134 on BMI, and of *BNDF* rs11030104 and *MC4R* rs12970134 on WC. Nevertheless, no interaction was found between PA and any SNP’s effect on obesity and central obesity. A meta-analysis and other studies also identified no SNP interaction with PA for WC_adjBMI_ or obesity [18,28]. This may be due to the bias inherent in self-reported estimates and measurement errors of PA [29]. Further studies should be done with relatively high accuracy and precisely measured PA to reveal the interactions of PA and SNPs in obesity and central obesity. Simultaneously, LTSB increased the effects of *FTO* rs9939609 and *FTO* rs8050136 on obesity and central obesity. So, engaging in PA and less LTSB could mitigate the impact of risk alleles on a genetic predisposition to obesity.

One of our novel findings was that *MC4R* rs12970134 interacted with dietary in BMI. High-energy dietary intake aggravated the influence of *MC4R* rs12970134 on an increased BMI. A study of the interaction between the *FTO* gene and dietary intake showed that the association between *FTO* and BMI was more pronounced in those with a dietary intake of high fat and low carbohydrates and fiber [30]. The inactive/high intake women had a 39.0% greater risk of obesity associated with each A allele in *FTO* carried when compared with the non-carriers [31]. Diet intervention could change the association between *FTO* and body-weight changes with a significant body-weight reduction [32]. Our results indicated that *FTO* rs8050136 increased the risk of obesity by 77% (OR = 1.77, 95%CI 1.20–2.62) among the participants with a high dietary energy intake. The specific mechanism of increasing energy intake that made *FTO* more pronounced in obesity is still unclear. One study found that the *FTO* gene was positively correlated with the percentage of energy derived from fat, and negatively correlated with the percentage of energy from carbohydrates [33], indicating that the association between the *FTO* gene and obesity may be regulated by energy intake. This conjecture needs to be confirmed in further studies.

To our knowledge, this is the first study that found that a high socioeconomic status aggravates the role of genes in obesity among Chinese adults. Some researchers believe that people of low social class are at a disadvantage in terms of economic level and education [34]. The gene–obesogenic environment interactions showed that a low socioeconomic position accentuated the risk of obesity in genetically susceptible adults [22]. A similar conclusion was reached in studies among European-American and African-American adolescents, as obesity-candidate genes carriers had a higher percentage of body fat with low socioeconomic status [35]. The above studies indicated that in developed countries, low socioeconomic status aggravated the genetic susceptibility to obesity. Contrary to the results in developed countries, our results indicate that high socioeconomic status aggravated the effect of SNPs on obesity. In developed countries, the people with a high socioeconomic status were more likely to choose a healthy lifestyle in developed countries. However, with rapid economic growth and changes in lifestyle in China, the people with a high socioeconomic status had more opportunities to choose food, and were more likely to eat high-energy food [36]. The people with a high socioeconomic status also were more likely to travel using transportation that lacked PA, such as private cars, taxis, motorcycles, etc. [37]. Research also showed that the family per capita annual income was positively correlated with obesity in China [38]. This might be why a high socioeconomic status aggravates the effect of genes on obesity in China.

The limitation of this study lies in its use of self-reported measurements, which could lead to spurious interaction. In addition, long-term changes in environmental factors, such as diet and exercise after birth, were unobtainable, which limited the ability to identify long-term genetic influences. The narrow age range also made it unclear whether this gene–environmental interaction occurred when younger, so further experiments will need to be done to reveal whether this kind of gene–environment effect occurs in younger people or elder adults.

## 5. Conclusions

In conclusion, a low level of PA, a high socioeconomic status, a long-time LTSB and a high dietary energy intake aggravated the predisposition of SNPs to BMI, WC, obesity and central obesity among Chinese adults. Our results reinforced that postnatal environment factors could change the influence of risk alleles on genetic predisposition to obesity. It was suggested that we should pay more attention to the influence of environmental factors on gene expression.

## Figures and Tables

**Table 1 genes-12-00270-t001:** Basic characteristics of the 2216 study subjects.

Characteristics	Total	Obesity	*p*	Central Obesity	*p*
Total	2216	295 (13.35%)		682 (30.9%)	
Age (year)	49.7 (48.7,51.3)	50.1 (48.8,51.4)	0.048	50.1 (48.8,51.4)	0.324
Gender (n, %)			0.072		0.006
Male	879 (39.7%)	102 (11.6%)		242 (27.5%)	
Female	1337 (60.3%)	193 (14.5%)		440 (33.0%)	
Education level (n, %)			0.951		0.460
Illiterate or primary *	787 (35.5%)	107 (13.6%)		232 (29.6%)	
Junior school	951 (42.9%)	126 (13.3%)		293 (30.9%)	
SHS and above **	478 (21.6%)	62 (13.0%)		157 (32.9%)	
Economic status (n, %)			0.745		0.961
Low	1146 (51.7%)	150 (13.1%)		349 (30.5%)	
Middle	834 (37.6%)	114 (13.7%)		260 (31.2%)	
High	157 (7.1%)	18 (11.5%)		47 (30.3%)	
No answer	79 (3.6%)	13 (16.5%)		26 (32.9%)	
Housework (n, %)			0.144		0.198
≤P_30_	653 (29.7%)	74 (11.4%)		190 (29.1%)	
P_30_–P_60_	474 (21.5%)	64 (13.5%)		139 (29.3%)	
≥P_60_	1074 (48.8%)	157 (14.7%)		350 (32.7%)	
Transportation modes (n, %)			0.228		0.506
Inactive	976 (44.3%)	140 (14.4%)		308 (31.7%)	
Active	1225 (55.7%)	155 (12.7%)		371 (30.3%)	
Physical activity (n, %) ^#^			0.107		0.076
No	1909 (86.7%)	247 (13.0%)		576 (30.2%)	
Yes	292 (13.3%)	48 (16.4%)		103 (35.4%)	
Sedentary behaviors (n, %)			0.124		0.965
No	392 (17.8%)	62 (15.9%)		121 (31.0%)	
Yes	1808 (82.2%)	233 (12.9%)		558 (30.9%)	
Energy intake/day (Kcal)	1379.8 (1051.1,1720.8)	1315.4 (1028.7, 1679.4)	0.884	1315.4 (1028.7,1679.4)	0.259
Energy intake/day (n, %)			0.693		0.286
≤P_25_	431 (25.0%)	63 (14.6%)		136 (31.6%)	
P_25_–P_50_	432 (25.0%)	51 (11.9%)		120 (27.8%)	
P_50_–P_75_	431 (25.0%)	57 (13.3%)		129 (29.9%)	
≥P_75_	432 (25.0%)	56 (13.0%)		145 (33.7%)	
BMI (kg/m^2^)	24.0 (21.9,26.4)	23.5 (21.6,25.4)	<0.001	29.7 (28.8,31.1)	<0.001
Waist circumference (cm)	82.0 (75.2,88.8)	95.5 (91.1,100.4)	<0.001	95.5 (91.1,100.4)	<0.001

* Illiterate or primary school; ** Senior high school and above; ^#^ Leisure time physical activity.

**Table 2 genes-12-00270-t002:** Factor analysis of obesity-related environmental factors.

Environmental Factors	Factor 1 *	Factor 2 *	Factor 3 *	Factor 4 *
Leisure-time physical activity	0.681			
Housework	0.639			
Transportation mode	0.618			
Education level		0.720		
Economic level		0.634		
Everyday energy intake			0.951	
Leisure-time sedentary behavior				0.974
Eigenvalues	1.497	1.171	1.000	0.964
Contribution rate (%)	0.214	0.167	0.143	0.138
Cumulative contribution rate of variance (%)	0.214	0.381	0.524	0.662

* Only displays factor-loading values > 0.45, which was considered as the principal component of the factor.

**Table 3 genes-12-00270-t003:** Interaction of genes and environment on BMI and waist circumference.

SNPs	Gene	Physical Activity		Socioeconomic Status		Dietary Energy Intake		Sedentary Behavior	
		β	*p*	β	*p*	β	*p*	β	*p*
**BMI**									
rs9939609	*FTO* **	0.082	0.320	0.203	0.010	0.077	0.314	0.145	0.062
rs11030104	*BDNF* **	0.083	0.316	0.213	0.007	0.088	0.253	0.135	0.079
rs6265	*BDNF* **	0.069	0.408	0.202	0.012	0.067	0.395	0.138	0.078
rs16892496	*TRHR* **	−0.014	0.854	0.103	0.162	−0.003	0.966	0.060	0.424
rs7832552	*TRHR* **	−0.131	0.099	−0.001	0.988	−0.098	0.183	−0.046	0.534
rs2568958	*1p31*	−0.132	0.438	0.343	0.015	−0.075	0.596	0.169	0.230
rs7561317	*TMEM18*	−0.137	0.434	0.340	0.020	−0.109	0.450	0.129	0.372
rs574367	*SEC16B*	0.089	0.224	0.195	0.005	0.090	0.197	0.140	0.044
rs12454712	*BCL2*	−0.071	0.400	0.090	0.269	−0.021	0.790	0.037	0.640
rs12970134	*MC4R*	−0.164	0.030	0.270	<0.001	0.140	0.049	0.214	0.003
rs8050136	*FTO* **	0.040	0.618	0.177	0.023	0.077	0.313	0.127	0.099
rs2237892	*KCNQ1*	−0.122	0.276	0.072	0.507	−0.055	0.613	−0.106	0.336
**WC ***									
rs9939609	*FTO* **	0.202	0.397	0.610	0.007	0.241	0.286	0.414	0.065
rs11030104	*BDNF* **	0.269	0.264	0.725	0.002	0.330	0.142	0.459	0.041
rs6265	*BDNF* **	0.245	0.312	0.681	0.004	0.287	0.207	0.439	0.052
rs16892496	*TRHR* **	−0.093	0.676	0.311	0.144	−0.009	0.966	0.138	0.525
rs7832552	*TRHR* **	−0.426	0.044	0.028	0.900	−0.279	0.195	−0.164	0.447
rs2568958	*1p31*	−0.227	0.648	1.301	0.002	0.098	0.810	0.670	0.102
rs7561317	*TMEM18*	−0.646	0.203	0.976	0.021	−0.294	0.483	0.281	0.503
rs574367	*SEC16B*	0.176	0.407	0.533	0.008	0.214	0.288	0.361	0.074
rs12454712	*BCL2*	−0.450	0.048	0.052	0.824	−0.269	0.245	−0.119	0.605
rs12970134	*MC4R*	0.370	0.091	0.737	<0.001	0.367	0.077	0.562	0.007
rs8050136	*FTO* **	0.131	0.578	0.552	0.014	0.214	0.331	0.343	0.124
rs2237892	*KCNQ1*	−0.317	0.327	0.281	0.369	−0.143	0.647	−0.351	0.267

* Waist circumference. ** The two SNPs on the same gene are in linkage disequilibrium.

**Table 4 genes-12-00270-t004:** The interaction between physical activity and SNPs in obesity and central obesity.

Environmental Factor *	SNPs	Obesity	OR (95%CI)	*p*	AP (%)	Central Obesity	OR (95%CI)	*p*	AP (%)
Physical activity	rs9939609				−54.57%				−15.19%
−	T	104 (12.7%)	1			267 (32.3%)	1		
+	T	99 (12.4%)	1.14 (0.80, 1.61)	0.464		228 (28.7%)	0.96 (0.75, 1.24)	0.762	
−	A	48 (19.5%)	1.67 (1.15, 2.44)	0.008		90 (36.6%)	1.21 (0.90, 1.62)	0.218	
+	A	27 (12.7%)	1.17 (0.71, 1.92)	0.532		63 (29.7%)	1.01 (0.71, 1.45)	0.939	
Physical activity	rs11030104				−25.18%				13.98%
−	G	26 (11.6%)	1			67 (29.9%)	1		
+	G	24 (11.8%)	1.19 (0.64, 2.23)	0.578		46 (22.6%)	0.79 (0.50, 1.25)	0.308	
−	A	122 (14.9%)	1.34 (0.85, 2.11)	0.203		278 (33.9%)	1.2 (0.87, 1.66)	0.257	
+	A	94 (12.1%)	1.23 (0.74, 2.03)	0.426		232 (29.9%)	1.15 (0.81, 1.65)	0.438	
Physical activity	rs6265				−29.51%				5.18%
−	T	25 (11.3%)	1			63 (28.4%)	1		
+	T	24 (12.4%)	1.29 (0.69, 2.43)	0.424		45 (23.2%)	0.85 (0.53, 1.36)	0.497	
−	C	125 (15.0%)	1.39 (0.88, 2.19)	0.162		288 (34.4%)	1.32 (0.96, 1.83)	0.093	
+	C	99 (12.6%)	1.30 (0.78, 2.15)	0.311		241 (30.6%)	1.24 (0.86, 1.77)	0.250	
Physical activity	rs16892496				0.93%				12.91%
−	A	36 (12.6%)	1			101 (35.1%)	1		
+	A	30 (11.4%)	1.02 (0.59, 1.78)	0.945		77 (29.2%)	0.86 (0.58, 1.26)	0.430	
−	C	113 (14.5%)	1.15 (0.77, 1.72)	0.491		253 (32.4%)	0.88 (0.66, 1.17)	0.380	
+	C	97 (13.0%)	1.18 (0.75, 1.86)	0.465		216 (29.0%)	0.84 (0.61, 1.16)	0.300	
Physical activity	rs7832552				28.97%				9.10%
−	T	42 (17.0%)	1			85 (34.4%)	1		
+	T	32 (12.6%)	0.84 (0.49, 1.43)	0.520		72 (28.5%)	0.87 (0.58, 1.30)	0.494	
−	C	109 (13.1%)	0.74 (0.50, 1.10)	0.133		274 (32.7%)	0.93 (0.69, 1.26)	0.638	
+	C	96 (12.5%)	0.82 (0.53, 1.27)	0.367		222 (28.8%)	0.88 (0.63, 1.23)	0.451	
Physical activity	rs12454712				−34.08%				−28.49%
−	C	25 (11.4%)	1			72 (32.9%)	1		
+	C	24 (13.1%)	1.34 (0.72, 2.50)	0.358		60 (32.8%)	1.11 (0.72, 1.73)	0.629	
−	T	130 (14.8%)	1.34 (0.85, 2.12)	0.209		292 (33.1%)	1.01 (0.74, 1.38)	0.958	
+	T	103 (12.3%)	1.25 (0.76, 2.07)	0.380		230 (27.5%)	0.87 (0.62, 1.24)	0.453	
Physical activity	rs12970134				6.25%				−2.87%
−	G	100 (14.0%)	1			233 (32.4%)	1		
+	G	83 (12.2%)	1.00 (0.69, 1.44)	0.985		192 (28.2%)	0.91 (0.70, 1.20)	0.517	
−	A	47 (14.3%)	1.02 (0.70, 1.49)	0.901		113 (34.2%)	1.09 (0.83, 1.44)	0.542	
+	A	40 (13.1%)	1.09 (0.70, 1.69)	0.708		90 (29.5%)	0.98 (0.70, 1.35)	0.886	
Physical activity	rs8050136				−90.90%				−20.85%
−	C	105 (12.5%)	1			272 (32.4%)	1		
+	C	103 (12.8%)	1.19 (0.84, 1.68)	0.319		230 (28.7%)	0.96 (0.75, 1.24)	0.759	
−	A	53 (19.9%)	1.74 (1.21, 2.50)	0.003		96 (36.1%)	1.18 (0.88, 1.58)	0.259	
+	A	24 (11.1%)	1.01 (0.61, 1.69)	0.966		61 (28.2%)	0.95 (0.66, 1.36)	0.759	
Physical activity	rs574367				13.42%				13.52%
−	G	95 (14.3%)	1			218 (32.8%)	1		
+	G	75 (12.0%)	0.95 (0.65, 1.38)	0.785		171 (27.3%)	0.87 (0.66, 1.15)	0.338	
−	T	54 (14.0%)	0.98 (0.68, 1.41)	0.911		133 (34.1%)	1.07 (0.82, 1.39)	0.632	
+	T	47 (13.2%)	1.07 (0.70, 1.64)	0.746		112 (31.7%)	1.09 (0.79, 1.49)	0.603	
Physical activity	Rs2237892				66.27%				16.44%
−	C	69 (14.7%)	1			156 (33.1%)	1		
+	C	53 (12.2%)	0.92 (0.58, 1.46)	0.725		127 (29.3%)	1.03 (0.74, 1.45)	0.850	
−	T	7 (7.3%)	0.45 (0.20, 1.01)	0.054		30 (31.3%)	0.91 (0.57, 1.46)	0.685	
+	T	16 (14%)	1.1 (0.57, 2.11)	0.773		35 (30.7%)	1.13 (0.69, 1.84)	0.637	

* Environmental factors were divided into binary variables with the P_50_ of the factor score as the cut-off value. Physical activity: “−“ = not taking part in physical activity often, “+” = taking part in physical activity often. The first subgroup was used as the reference group.

**Table 5 genes-12-00270-t005:** The interaction between socioeconomic status and SNPs in obesity and central obesity.

Environmental Factor *	SNPs	Obesity	OR (95%CI)	*p*	AP (%)	Central Obesity	OR (95%CI)	*p*	AP (%)
Socioeconomic status	rs9939609				−69.21%				−34.86%
−	T	88 (11.2%)	1			224 (28.4%)	1		
+	T	115 (13.9%)	1.34 (0.99, 1.81)	0.055		271 (32.6%)	1.28 (1.03, 1.59)	0.024	
−	A	46 (19.0%)	1.85 (1.25, 2.74)	0.002		85 (35.1%)	1.36 (1.00, 1.85)	0.048	
+	A	29 (13.4%)	1.30 (0.83, 2.04)	0.261		68 (31.5%)	1.22 (0.88, 1.69)	0.237	
Socioeconomic status	rs11030104				−1.63%				−15.89%
−	G	23 (11.1%)	1			47 (22.6%)	1		
+	G	27 (12.3%)	1.16 (0.64, 2.10)	0.625		66 (30.0%)	1.52 (0.98, 2.34)	0.062	
−	A	106 (13.3%)	1.22 (0.75, 1.97)	0.424		248 (31.0%)	1.53 (1.07, 2.19)	0.021	
+	A	110 (13.8%)	1.35 (0.84, 2.19)	0.217		262 (32.9%)	1.76 (1.23, 2.52)	0.002	
Socioeconomic status	rs6265				20.31%				−11.39%
−	T	25 (12.4%)	1			45 (22.3%)	1		
+	T	24 (11.2%)	0.91 (0.50, 1.65)	0.756		63 (29.4%)	1.48 (0.95, 2.31)	0.082	
−	C	108 (13.3%)	1.06 (0.67, 1.69)	0.798		255 (31.3%)	1.57 (1.09, 2.26)	0.015	
+	C	116 (14.3%)	1.22 (0.77, 1.94)	0.400		274 (33.8%)	1.84 (1.28, 2.65)	0.001	
Socioeconomic status	rs16892496				−19.69%				−40.87%
−	A	30 (10.8%)	1			78 (27.7%)	1		
+	A	36 (13.4%)	1.33 (0.79, 2.23)	0.283		100 (37.0%)	1.59 (1.11, 2.29)	0.011	
−	C	104 (13.8%)	1.31 (0.85, 2.01)	0.225		229 (30.4%)	1.13 (0.83, 1.53)	0.443	
+	C	106 (13.7%)	1.37 (0.89, 2.10)	0.158		240 (31.0%)	1.22 (0.90, 1.66)	0.197	
Socioeconomic status	rs7832552				13.87%				33.49%
−	T	36 (15.3%)	1			78 (33.3%)	1		
+	T	38 (14.3%)	0.99 (0.60, 1.64)	0.982		79 (29.7%)	0.90 (0.62, 1.32)	0.599	
−	C	100 (12.2%)	0.78 (0.52, 1.18)	0.243		233 (28.4%)	0.80 (0.59, 1.09)	0.160	
+	C	105 (13.3%)	0.90 (0.60, 1.36)	0.622		263 (33.4%)	1.06 (0.77, 1.44)	0.728	
Socio-economic status	rs2568958				−136.22%				−30.68%
−	G	1 (11.1%)	1			1 (11.1%)	1		
+	G	2 (25.0%)	2.94 (0.21, 40.86)	0.421		2 (25.0%)	2.84 (0.21, 39.36)	0.436	
−	A	133 (12.9%)	1.19 (0.15, 9.68)	0.868		308 (29.8%)	3.32 (0.41, 26.76)	0.260	
+	A	141 (13.6%)	1.33 (0.16, 10.77)	0.790		338 (32.6%)	3.95 (0.49, 31.84)	0.197	
Socioeconomic status	rs7561317				95.64%				342.70%
−	A	1 (25.0%)	1			3 (75.0%)	1		
+	A	1 (12.5%)	0.52 (0.02, 11.53)	0.682		4 (50.0%)	0.40 (0.03, 5.65)	0.495	
−	G	130 (12.9%)	0.50 (0.05, 4.85)	0.550		299 (29.6%)	0.16 (0.02, 1.50)	0.107	
+	G	137 (13.4%)	0.55 (0.06, 5.36)	0.607		331 (32.3%)	0.19 (0.02, 1.79)	0.145	
Socioeconomic status	rs574367				2.28%				2.74%
−	G	83 (12.8%)	1			186 (28.7%)	1		
+	G	87 (13.6%)	1.12 (0.81, 1.56)	0.482		203 (31.6%)	1.21 (0.95, 1.53)	0.127	
−	T	47 (13.1%)	1.03 (0.70, 1.52)	0.868		113 (31.3%)	1.14 (0.86, 1.51)	0.357	
+	T	54 (14.1%)	1.18 (0.82, 1.72)	0.373		132 (34.6%)	1.39 (1.05, 1.82)	0.020	
Socioeconomic status	rs12454712				−39.84%				−50.65%
−	C	18 (9.5%)	1			51 (26.7%)	1		
+	C	31 (14.6%)	1.72 (0.93, 3.20)	0.086		81 (38.4%)	1.79 (1.17, 2.74)	0.008	
−	T	117 (13.6%)	1.51 (0.89, 2.55)	0.124		258 (29.9%)	1.18 (0.83, 1.68)	0.365	
+	T	116 (13.6%)	1.59 (0.94, 2.70)	0.083		264 (30.9%)	1.30 (0.91, 1.86)	0.144	
Socioeconomic status	rs12970134				−6.63%				−0.30%
−	G	85 (12.4%)	1			197 (28.7%)	1		
+	G	98 (13.7%)	1.19 (0.87, 1.63)	0.287		228 (31.9%)	1.23 (0.97, 1.55)	0.082	
−	A	43 (13.4%)	1.10 (0.74, 1.63)	0.636		97 (30.4%)	1.10 (0.82, 1.46)	0.543	
+	A	44 (14.0%)	1.21 (0.81, 1.79)	0.350		106 (33.5%)	1.32 (0.99, 1.76)	0.061	
Socioeconomic status	rs8050136				−68.42%				−15.56%
−	C	90 (11.4%)	1				1		
+	C	118 (14.0%)	1.33 (0.99, 1.79)	0.060		228 (28.7%)	1.25 (1.01, 1.55)	0.04	
−	A	48 (18.5%)	1.76 (1.20, 2.58)	0.004		274 (32.4%)	1.21 (0.89, 1.63)	0.226	
+	A	29 (13.1%)	1.24 (0.79, 1.94)	0.354		85 (32.7%)	1.26 (0.91, 1.74)	0.161	
Socioeconomic status	Rs2237892				44.03%				39.44%
−	C	52 (11.6%)	1			131 (29.2%)	1		
+	C	70 (15.4%)	1.46 (0.99, 2.16)	0.057		152 (33.4%)	1.3 (0.98, 1.73)	0.071	
−	T	6 (5.6%)	0.45 (0.19, 1.08)	0.075		25 (23.2%)	0.74 (0.45, 1.21)	0.232	
+	T	17 (16.7%)	1.63 (0.89, 2.98)	0.111		40 (39.2%)	1.72 (1.09, 2.71)	0.019	

* Environmental factors were divided into binary variables with the P_50_ of the factor score as the cut-off value. Socioeconomic status: “−“ = low socioeconomic status, “+” = high socioeconomic status. The first subgroup was used as the reference group.

**Table 6 genes-12-00270-t006:** The interaction between dietary energy intake and SNPs in obesity and central obesity.

Environmental Factor *	SNPs	Obesity	OR (95%CI)	*p*	AP (%)	Central Obesity	OR (95%CI)	*p*	AP (%)
Dietary energy intake	rs11030104				2.80%				10.71%
−	G	24 (10.3%)	1			64 (27.5%)	1		
+	G	26 (13.3%)	1.32 (0.73, 2.38)	0.359		49 (25.1%)	0.87 (0.56, 1.34)	0.522	
−	A	92 (11.8%)	1.17 (0.73, 1.89)	0.509		248 (31.6%)	1.23 (0.89, 1.71)	0.210	
+	A	124 (15.3%)	1.54 (0.97, 2.45)	0.070		262 (32.4%)	1.23 (0.89, 1.70)	0.207	
Dietary energy intake	rs6265				9.71%				2.87%
−	T	25 (11.1%)	1			60 (26.7%)	1		
+	T	24 (12.6%)	1.13 (0.62, 2.06)	0.682		48 (25.1%)	0.91 (0.58, 1.41)	0.664	
−	C	98 (12.3%)	1.13 (0.71, 1.80)	0.606		261 (32.7%)	1.35 (0.97, 1.87)	0.079	
+	C	126 (15.2%)	1.40 (0.89, 2.21)	0.150		268 (32.4%)	1.29 (0.93, 1.79)	0.132	
Dietary energy intake	rs16892496				0.13%				−1.79%
−	A	27 (10.3%)	1			85 (32.2%)	1		
+	A	39 (13.6%)	1.32 (0.78, 2.23)	0.298		93 (32.3%)	0.98 (0.68, 1.40)	0.900	
−	C	95 (12.2%)	1.19 (0.76, 1.88)	0.446		240 (30.8%)	0.94 (0.69, 1.26)	0.663	
+	C	115 (15.3%)	1.52 (0.97, 2.37)	0.068		229 (30.6%)	0.90 (0.66, 1.21)	0.478	
Dietary energy intake	rs7832552				13.84%				−2.53%
−	T	37 (13.9%)	1			83 (31.2%)	1		
+	T	37 (15.7%)	1.14 (0.69, 1.86)	0.616		74 (31.6%)	0.99 (0.68, 1.45)	0.964	
−	C	85 (10.8%)	0.76 (0.50, 1.14)	0.183		243 (30.8%)	0.98 (0.73, 1.33)	0.909	
+	C	120 (14.6%)	1.03 (0.69, 1.54)	0.875		253 (30.9%)	0.95 (0.70, 1.28)	0.737	
Dietary energy intake	rs2568958				−174.49%				−81.05%
−	G	1 (10.0%)	1			1 (10.0%)	1		
+	G	2 (28.6%)	4.04 (0.29, 56.87)	0.301		2 (28.6%)	4.07 (0.29, 57.18)	0.297	
−	A	123 (11.9%)	1.28 (0.16, 10.20)	0.818		326 (31.4%)	4.27 (0.54, 33.90)	0.169	
+	A	151 (14.6%)	1.57 (0.20, 12.54)	0.669		320 (31.0%)	4.06 (0.51, 32.20)	0.185	
Dietary energy intake	rs574367				22.64%				10.14%
−	G	79 (12.2%)	1			200 (30.8%)	1		
+	G	91 (14.2%)	1.16 (0.84, 1.60)	0.379		189 (29.4%)	0.91 (0.72, 1.15)	0.433	
−	T	41 (11.0%)	0.90 (0.60, 1.34)	0.604		121 (32.4%)	1.08 (0.82, 1.43)	0.566	
+	T	60 (16.3%)	1.37 (0.95, 1.97)	0.095		124 (33.6%)	1.11 (0.84, 1.45)	0.474	
Dietary energy intake	rs12454712				−16.26%				−2.11%
−	C	19 (9.5%)	1			65 (32.3%)	1		
+	C	30 (14.9%)	1.63 (0.88, 3.00)	0.119		67 (33.3%)	1.01 (0.67, 1.53)	0.963	
−	T	104 (12.0%)	1.31 (0.78, 2.19)	0.305		262 (30.2%)	0.91 (0.65, 1.27)	0.577	
+	T	129 (15.2%)	1.67 (1.00, 2.78)	0.049		260 (30.6%)	0.90 (0.65, 1.25)	0.535	
Dietary energy intake	rs12970134				−4.38%				−9.15%
−	G	83 (11.7%)	1			215 (30.3%)	1		
+	G	100 (14.5%)	1.23 (0.90, 1.68)	0.203		210 (30.4%)	0.97 (0.77, 1.22)	0.795	
−	A	40 (12.7%)	1.09 (0.73, 1.63)	0.689		104 (32.9%)	1.13 (0.85, 1.50)	0.408	
+	A	47 (14.7%)	1.26 (0.85, 1.85)	0.247		99 (31.0%)	1.01 (0.76, 1.34)	0.969	
Dietary energy intake	rs8050136				19.84%				−14.98%
−	C	94 (11.3%)	1			251 (30.0%)	1		
+	C	114 (14.1%)	1.25 (0.93, 1.68)	0.138		251 (31.2%)	1.02 (0.82, 1.26)	0.878	
−	A	32 (13.2%)	1.17 (0.76, 1.80)	0.471		82 (33.7%)	1.17 (0.87, 1.59)	0.305	
+	A	45 (18.8%)	1.77 (1.20, 2.62)	0.004		75 (31.4%)	1.04 (0.76, 1.41)	0.828	
Dietary energy intake	Rs2237892				−10.66%				16.6%
−	C	45 (10.1%)	1			133 (29.8%)			
+	C	77 (16.8%)	1.78 (1.20, 2.63)	0.004		150 (32.8%)	1.12 (0.84, 1.48)	0.447	
−	T	9 (8.2%)	0.80 (0.38, 1.70)	0.565		30 (27.3%)	0.91 (0.57, 1.45)	0.689	
+	T	14 (14%)	1.43 (0.75, 2.72)	0.281		35 (35%)	1.23 (0.78, 1.95)	0.379	

* Environmental factors were divided into binary variables with the P_50_ of the factor score as the cut-off value. Dietary energy intake: “−“ = low dietary energy intake, “+” = high dietary energy intake. The first subgroup was used as the reference group.

**Table 7 genes-12-00270-t007:** The interaction between sedentary behavior and SNPs in obesity and central obesity.

Environmental Factor *	SNPs	Obesity	OR (95%CI)	*p*	AP (%)	Central Obesity	OR (95%CI)	*p*	AP (%)
Sedentary behavior	rs9939609				2.88%				21.62%
−	T	93 (11.6%)	1			234 (29.1%)	1		
+	T	110 (13.5%)	1.21 (0.90, 1.63)	0.207		261 (32.0%)	1.16 (0.94, 1.43)	0.171	
−	A	35 (15.1%)	1.37 (0.90, 2.09)	0.140		67 (28.9%)	1.00 (0.73, 1.39)	0.981	
+	A	40 (17.7%)	1.63 (1.09, 2.45)	0.018		86 (38.1%)	1.49 (1.09, 2.02)	0.012	
Sedentary behavior	rs11030104				12.31%				−12.31%
−	G	26 (11.4%)	1			52 (22.8%)	1		
+	G	24 (12.0%)	1.05 (0.58, 1.89)	0.881		61 (30.5%)	1.48 (0.96, 2.28)	0.078	
−	A	98 (12.6%)	1.11 (0.70, 1.76)	0.663		235 (30.2%)	1.46 (1.03, 2.06)	0.033	
+	A	118 (14.5%)	1.32 (0.84, 2.07)	0.236		275 (33.7%)	1.72 (1.22, 2.43)	0.002	
Sedentary behavior	rs6265				11.71%				−10.57%
−	T	26 (11.4%)	1			53 (23.1%)	1		
+	T	23 (12.3%)	1.09 (0.60, 1.98)	0.779		55 (29.4%)	1.38 (0.89, 2.15)	0.149	
−	C	100 (12.6%)	1.12 (0.71, 1.77)	0.637		247 (31.2%)	1.50 (1.07, 2.11)	0.020	
+	C	124 (14.9%)	1.37 (0.87, 2.15)	0.175		282 (33.9%)	1.70 (1.21, 2.39)	0.002	
Sedentary behavior	rs16892496				32.91%				0.52%
−	A	33 (12.7%)	1			79 (30.2%)	1		
+	A	33 (11.5%)	0.91 (0.54, 1.52)	0.706		99 (34.1%)	1.22 (0.85, 1.74)	0.285	
−	C	94 (12.0%)	0.94 (0.62, 1.44)	0.789		222 (28.5%)	0.92 (0.68, 1.26)	0.613	
+	C	116 (15.5%)	1.27 (0.84, 1.92)	0.264		247 (33.0%)	1.15 (0.85, 1.56)	0.381	
Sedentary behavior	rs7832552				−0.61%				−9.03%
−	T	36 (13.9%)	1			74 (28.5%)	1		
+	T	38 (15.8%)	1.16 (0.71, 1.91)	0.553		83 (34.6%)	1.32 (0.90, 1.93)	0.154	
−	C	95 (11.9%)	0.83 (0.55, 1.25)	0.375		232 (28.9%)	1.01 (0.74, 1.38)	0.948	
+	C	110 (13.7%)	0.99 (0.66, 1.48)	0.943		264 (32.8%)	1.22 (0.90, 1.66)	0.209	
Sedentary behavior	rs2568958				219.79%				8.00%
−	G	2 (28.6%)	1			1 (14.3%)	1		
+	G	1 (10.0%)	0.24 (0.02, 3.36)	0.288		2 (20.0%)	1.32 (0.10, 18.23)	0.838	
−	A	125 (12.1%)	0.31 (0.06, 1.62)	0.165		299 (28.9%)	2.19 (0.26, 18.36)	0.470	
+	A	149 (14.4%)	0.38 (0.07, 1.99)	0.252		347 (33.5%)	2.73 (0.33, 22.82)	0.355	
Sedentary behavior	rs7561317				203.14%				137.45%
−	A	1 (33.3%)	1			2 (66.7%)	1		
+	A	1 (11.1%)	0.30 (0.01, 7.13)	0.452		5 (55.6%)	0.68 (0.04, 10.63)	0.785	
−	G	122 (12.0%)	0.31 (0.03, 3.49)	0.344		293 (28.8%)	0.22 (0.02, 2.43)	0.215	
+	G	145 (14.2%)	0.38 (0.03, 4.28)	0.435		337 (33.0%)	0.27 (0.02, 2.97)	0.283	
Sedentary behavior	rs574367				−25.42%				1.23%
−	G	75 (11.7%)	1			180 (28.1%)	1		
+	G	95 (14.6%)	1.30 (0.94, 1.80)	0.117		209 (32.2%)	1.21 (0.96, 1.54)	0.114	
−	T	51 (13.7%)	1.20 (0.82, 1.76)	0.344		116 (31.1%)	1.16 (0.87, 1.53)	0.308	
+	T	50 (13.5%)	1.20 (0.82, 1.76)	0.361		129 (34.9%)	1.39 (1.05, 1.83)	0.020	
Sedentary behavior	rs12454712				−0.07%				19.96%
−	C	23 (11.2%)	1			67 (32.5%)	1		
+	C	26 (13.2%)	1.23 (0.67, 2.24)	0.504		65 (33.2%)	1.04 (0.69, 1.58)	0.857	
−	T	108 (12.6%)	1.15 (0.71, 1.86)	0.567		237 (27.6%)	0.80 (0.58, 1.11)	0.182	
+	T	125 (14.6%)	1.38 (0.86, 2.21)	0.187		285 (33.2%)	1.05 (0.76, 1.45)	0.781	
Sedentary behavior	rs12970134				−43.03%				−4.32%
−	G	78 (11.4%)	1			191 (27.8%)	1		
+	G	105 (14.7%)	1.36 (0.99, 1.86)	0.056		234 (32.8%)	1.28 (1.02, 1.61)	0.037	
−	A	47 (14.4%)	1.31 (0.89, 1.94)	0.171		98 (30.1%)	1.12 (0.84, 1.50)	0.430	
+	A	40 (13.0%)	1.17 (0.78, 1.76)	0.454		105 (34.0%)	1.34 (1.01, 1.79)	0.045	
Sedentary behavior	rs8050136				1.59%				20.82%
−	C	95 (11.7%)	1			237 (29.1%)	1		
+	C	113 (13.7%)	1.22 (0.91, 1.63)	0.186		265 (32.0%)	1.16 (0.94, 1.44)	0.164	
−	A	38 (14.8%)	1.33 (0.89, 2.00)	0.170		73 (28.4%)	0.98 (0.72, 1.34)	0.895	
+	A	39 (17.3%)	1.57 (1.05, 2.36)	0.029		84 (37.3%)	1.44 (1.06, 1.97)	0.021	
Sedentary behavior	Rs2237892				−1.93%				13.94%
−	C	51 (12.1%)	1			128 (30.4%)	1		
+	C	71 (14.7%)	1.26 (0.86, 1.86)	0.24		155 (32.1%)	1.10 (0.83, 1.46)	0.513	
−	T	11 (9.7%)	0.79 (0.4, 1.58)	0.51		32 (28.3%)	0.93 (0.59, 1.48)	0.766	
+	T	12 (12.4%)	1.03 (0.53, 2.03)	0.922		33 (34%)	1.20 (0.75, 1.92)	0.451	

* Environmental factors were divided into binary variables with the P_50_ of the factor score as the cut-off value. Sedentary behavior: “−“ = not engaging in sedentary behavior often, “+” = engaging in sedentary behavior often. The first subgroup was used as the reference group.

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
