# Peer review of "Effects of Gene-Environment Interaction on Obesity among Chinese Adults Born in the Early 1960s"

_genes, 2021, doi:10.3390/genes12020270_

Round 1

Reviewer 1 Report

The manuscript: “Effects of gene-environment interaction on obesity 2 among Chinese adults born in the early 1960s” aims to analyze the interaction of SNPs and environmental factors on BMI and waist circumference in the Chinese population. Its innovation is to focus only on a specific population of Chinese.
For a better comprehension of the phenomena of gene-environment interaction, I suggest to the author the following papers:
- Genome-wide physical activity interactions in adiposity - A meta-analysis of 200,452 adults; Graff et al. ; PLoS Genet. 2017 Apr 27;13(4):e1006528

- FTO genotype is associated with phenotypic variability of body mass index; Nature. 2012 October 11; 490(7419): 267–272. doi:10.1038/nature11401.

Here a few observations and questions:

2.2. Genotyping and SNP Selection section à Did you calculated the Cryptic relatedness of individuals? For example, rs12970134 has a great significance for Socioeconomic status, could the value biased by relatedness?

2.4. Statistical Analysis section:

Line 111, please provide a preference for the Kaiser Meyer Olkin test; KMO values less than 0.6 indicate the sampling is not adequate and that remedial action should be taken. Please explain some details of your own judgment for values between 0.5 and 0.6.

Lines 122-123: “It is often assumed that the odds ratio (OR) can be used instead of the relative risk.”; please add, at least, a reference

Table 3. for completeness I would suggest adding a column with gene (and -between brachets- chromosome) for each SNP shown; and in case of SNPs on the same chromosome, to indicate also the LD.

For example: the two SNPs for FTO are in LD: rs8050136(A) allele is correlated with rs9939609(A) allele and rs8050136(C) allele is correlated with rs9939609(T) allele.

Author Response

Dear reviewer,

We are extremely grateful for your positive and constructive feedback. The incorporation of your suggestions has strengthened the manuscript. According to your advice, we revised the relevant part, and highlighted that in manuscript. Here below is our description on revision. 

If you have any question, please contact us without hesitate.

Best regards,

Weiyan Gong

Description on revision:

Comment:  2.2. Genotyping and SNP Selection section à Did you calculated the Cryptic relatedness of individuals? For example, rs12970134 has a great significance for Socioeconomic status, could the value biased by relatedness?

Response: Thank you for your suggestion, all the participants we selected have no Cryptic relatedness. Page 3 line 18. Please see the details as follow:

All the participants had no cryptic relatedness.

2.4. Statistical Analysis section:

Line 111, please provide a preference for the Kaiser Meyer Olkin test; KMO values less than 0.6 indicate the sampling is not adequate and that remedial action should be taken. Please explain some details of your own judgment for values between 0.5 and 0.6.

Response: Thank you for your suggestion, exactly as you said, it’s better for factor analysis when KMO ≥ 0.6. There were also studies using KMO ≥ 0.5 when doing factor analyses, and one study was about the analyzing the gene-environment interaction on obesity (reference: Interaction between FTO gene polymorphism and 1ife style may contribute to obesity in Kazakh schoolchildren), so we use KMO ≥ 0.5 in this study. We added the references and explanation. Please see the details as follows:

Factor analysis generally can be done when KMO ≥ 0.5 and Bartlett's spherical test p < 0.05, while KMO≥ 0.6 would be more suitable[15,25]. In this study, we adopted KMO≥ 0.5 which was used in a previous study[15]. (Page 3 line 114-115)

 Lines 122-123: “It is often assumed that the odds ratio (OR) can be used instead of the relative risk.”; please add, at least, a reference

Response: Thank you for your suggestion. I am very sorry for the wrong expression. I want to express that ‘ Odds ration (OR) is generally used to assess risk’. (Page 3 lines 124-125)

Table 3. for completeness I would suggest adding a column with gene (and -between brachets- chromosome) for each SNP shown; and in case of SNPs on the same chromosome, to indicate also the LD.

For example: the two SNPs for FTO are in LD: rs8050136(A) allele is correlated with rs9939609(A) allele and rs8050136(C) allele is correlated with rs9939609(T) allele.

Response: Thank you for your suggestion, we added a column with gene, and the indicated the LD when the SNPs on the same gene(Page 5 line159-160). You can also see the details in the following table:

Table 3. Interaction of gene-environment on BMI and waist circumference.

SNPs

Gene

Physical activity

Socio-economic status

Dietary energy intake

Sedentary behavior

β

p

β

p

β

p

β

p

BMI

rs9939609

FTO**

0.082

0.320

0.203

0.010

0.077

0.314

0.145

0.062

rs11030104

BDNF**

0.083

0.316

0.213

0.007

0.088

0.253

0.135

0.079

rs6265

BDNF**

0.069

0.408

0.202

0.012

0.067

0.395

0.138

0.078

rs16892496

TRHR**

-0.014

0.854

0.103

0.162

-0.003

0.966

0.060

0.424

rs7832552

TRHR**

-0.131

0.099

-0.001

0.988

-0.098

0.183

-0.046

0.534

rs2568958

1p31

-0.132

0.438

0.343

0.015

-0.075

0.596

0.169

0.230

rs7561317

TMEM18

-0.137

0.434

0.340

0.020

-0.109

0.450

0.129

0.372

rs574367

SEC16B

0.089

0.224

0.195

0.005

0.090

0.197

0.140

0.044

rs12454712

BCL2

-0.071

0.400

0.090

0.269

-0.021

0.790

0.037

0.640

rs12970134

MC4R

-0.164

0.030

0.270

<0.001

0.140

0.049

0.214

0.003

rs8050136

FTO**

0.040

0.618

0.177

0.023

0.077

0.313

0.127

0.099

rs2237892

KCNQ1

-0.122

0.276

0.072

0.507

-0.055

0.613

-0.106

0.336

WC*

rs9939609

FTO**

0.202

0.397

0.610

0.007

0.241

0.286

0.414

0.065

rs11030104

BDNF**

0.269

0.264

0.725

0.002

0.330

0.142

0.459

0.041

rs6265

BDNF**

0.245

0.312

0.681

0.004

0.287

0.207

0.439

0.052

rs16892496

TRHR**

-0.093

0.676

0.311

0.144

-0.009

0.966

0.138

0.525

rs7832552

TRHR**

-0.426

0.044

0.028

0.900

-0.279

0.195

-0.164

0.447

rs2568958

1p31

-0.227

0.648

1.301

0.002

0.098

0.810

0.670

0.102

rs7561317

TMEM18

-0.646

0.203

0.976

0.021

-0.294

0.483

0.281

0.503

rs574367

SEC16B

0.176

0.407

0.533

0.008

0.214

0.288

0.361

0.074

rs12454712

BCL2

-0.450

0.048

0.052

0.824

-0.269

0.245

-0.119

0.605

rs12970134

MC4R

0.370

0.091

0.737

<0.001

0.367

0.077

0.562

0.007

rs8050136

FTO**

0.131

0.578

0.552

0.014

0.214

0.331

0.343

0.124

rs2237892

KCNQ1

-0.317

0.327

0.281

0.369

-0.143

0.647

-0.351

0.267

* Waist circumference.**The two SNPs on the same gene are in linkage disequilibrium.

Reviewer 2 Report

A very nice job on the gene-environment interaction research field. Here are some suggestions:

The first appeared abbreviation needs to include the full name, like the “BMI” in the abstract.

The paper proposes that the Interaction between genes and PA has few effects on WC, obesity, and central obesity. This kind of result is contrary to the conclusions of many previous papers. Because PA was considered as an important factor for WC or obesity in many types of research. More discussion is needed to explain the differences.

The subjects used in this paper were born in 1960, 1961, and 1963 and the average age of subjects was 49.7 years. To avoid the bias of the data, further experiments are needed to reveal whether this kind of gene-environment effect happens in younger or elder adults. Or review related researches of the different age range to compare with the results in this paper.

Some statistical analysis like mediation analysis may be more suitable for the Factor-gene-obesity data than the linear regression.

Author Response

Dear reviewer,

We are extremely grateful for your positive and constructive feedback. The incorporation of your suggestions has strengthened the manuscript. According to your advice, we revised the relevant part, and highlighted that in manuscript. Here below is our description on revision. 

If you have any question, please contact us without hesitate.

Best regards,

Weiyan Gong

Description on revision:

Comment: The first appeared abbreviation needs to include the full name, like the “BMI” in the abstract.

Response: According to your suggestion, we added the full name when the first appeared abbreviation, page 1 line 19 & 20, page 2 line75, page3 line 99.

Comment: The paper proposes that the Interaction between genes and PA has few effects on WC, obesity, and central obesity. This kind of result is contrary to the conclusions of many previous papers. Because PA was considered as an important factor for WC or obesity in many types of research. More discussion is needed to explain the differences.

Response: According to your suggestion, we added more discussion as follows:

Nevertheless, no interaction was found between PA and any SNPs effect on obesity and central obesity. A meta-analysis and other study also identified no SNPs interaction with PA for WC or obesity[18,28]. This may be due to the bias inherent in self-report estimates and measurement error of PA[29]. Further studies should be done with relatively high accurately and precisely measured PA to reveal the interactions of PA and SNPs on obesity and central obesity.  (page 6 line 197-202).

Comment: The subjects used in this paper were born in 1960, 1961, and 1963 and the average age of subjects was 49.7 years. To avoid the bias of the data, further experiments are needed to reveal whether this kind of gene-environment effect happens in younger or elder adults. Or review related researches of the different age range to compare with the results in this paper.

Response: Very good suggestion. In the future, we will do more experiments in different age groups to reveal whether this kind of gene-environment effect happens in younger or elder adults. We added the limitation of this study as follows:

Besides, the narrow age range made it unclear whether this gene-environmental interaction happened in younger, further experiments will be done to reveal whether this kind of gene-environment effect happens in younger or elder adults. (page 7 lines 238-241)

Comment: Some statistical analysis like mediation analysis may be more suitable for the Factor-gene-obesity data than the linear regression.

Response: Thank you for your suggestion. We did find that in addition to linear regression, some studies used mediation analysis to analyze the factor-gene-obesity data. In the future research, we will use the mediation analysis to analyze the factor-gene-obesity data.
